# Informal Community Growing Characteristics and the Satisfaction of Concerned Residents in Mountainous Urban Areas of Southwest China

**DOI:** 10.3390/ijerph192215178

**Published:** 2022-11-17

**Authors:** Zhong Xing, Canhui Cheng, Qiao Yu, Junyue Yang, Hao Ma, Jian Yang, Xiaomin Du

**Affiliations:** 1Faculty of Architecture and Urban Planning, Chongqing University, Chongqing 400045, China; 2Faculty of Architecture and Urban Planning, Chongqing Jiaotong University, Chongqing 400074, China; 3Faculty of Architecture and Urban Planning, Guizhou University, Guiyang 550025, China; 4China Aero Geophysical Survey and Remote Sensing Center for Natural Resources, Beijing 100083, China

**Keywords:** characteristics, informal community growing, reasons for growing, resident satisfaction, the urban areas of southwest China

## Abstract

Due to the mountainous terrain in the urban areas of southwest China, there are a large number of barren slopes in the community unsuitable for construction. These areas, alongside other unusable space which is often cultivated by residents to create informal community vegetable gardens and fruit growing areas, have become a “gray area” for urban management. This paper attempts to study the characteristics of informal community growing, the composition of growers, the motivation for growing, and the satisfaction of residents in urban areas in mountainous southwest China to explore its relative value. The sample area for the study was Yongchuan, Chongqing, Southwest China. Through a field survey, a semantic differential questionnaire, and data analysis, we found that: (1) growers use traditional cultivation methods to grow diverse fruits and vegetables according to the size of the slope, and the scale is so large that it serves as a local food supply; (2) growers are mainly vulnerable groups who use the land for economic gain and green food acquisition; and (3) growers and non-growers are more satisfied with the food supply and economic benefits generated by cultivation, while they are dissatisfied with the environmental and social benefits and the planting process. Satisfaction also varies with age, occupation, income, education, household registration, and farming experience. Based on the findings, this paper presents recommendations for the future transformation and development of informal community cultivation in mountainous areas. The study has implications for the construction of community gardens and urban agriculture in the mountains.

## 1. Introduction

The term “urban informality” originated in the 1960s in the U.S. [1,2] and involves the utilization of urban space by urban residents at their own will, ignoring the relevant laws. With rapid urbanization, more and more informal activities, such as informal housing, informal economy, informal community growing, etc., are gradually emerging in cities. According to statistics, approximately 1.8 billion people globally are engaged in informal activities [3], of which 114 million people are informally employed in China [4]. The 2009 UN-Habitat report on sustainable urban development identified informality and poverty, climate change, and resource scarcity as major challenges for cities [5], which have gradually drawn the attention of various disciplines such as planning, landscaping, and geography. As a result, informality is becoming increasingly important in the field of future urban development.

Informal behavior is prevalent in Chinese cities, where informal community planting is gradually becoming a topic of contemporary concern [6]. It refers to the spontaneous use of vacant and abandoned land (an irregular, small area of corner space) to cultivate fruits and vegetables for daily consumption without the permission of the community committee [7]. Rather than improving the community environment, the main purpose of the planting is to obtain food. Consequently, conflicts and environmental damages may occur [6,8]. The reason for this phenomenon is related to China’s urbanization in recent decades. From 2010 to 2020, the Chinese population urbanization rate increased from 49.68% to 63.89% [9]. Rapid urbanization over the past few decades has led to the continuous expansion of cities occupying the surrounding rural land, resulting in an influx of rural people into the cities. These individuals retain their rural farming habits and lifestyle, and reclaim abandoned urban space for cultivation. The phenomenon is prevalent in nearly all cities in China [10,11].

Informal community growing in China resembles community gardens in Europe and the United States. Community gardens are public lands in and around urban settlements that are planted and maintained by groups of residents, also known as community vegetable plots, where flowers, fruits, and vegetables can be grown, or any plot that provides food for the city. Research confirms that growing activities in community gardens are beneficial in improving residents’ mental and physical health [12,13,14,15] and social interaction [13,16]. Meanwhile, community gardens also play a role in recreation and leisure, providing green agricultural products [17,18], simulating community vitality [19,20], and promoting sustainable community development [21]. Therefore, the construction of community gardens has become popular in many countries and regions [22], including the United States [23,24,25,26], Canada [27,28], the United Kingdom [29,30,31], Spain [32], Australia [33], Israel [34], Singapore [35], etc. Informal community growing in China is equivalent to a preliminary version of community gardens, where positive and negative effects coexist [6]. The gardens are also vulnerable to official purges due to significant negative effects [36]. Does informal community growing in China have the potential value of incorporating multi-purpose community gardens into urban agricultural systems? This topic deserves in-depth study.

Many early community gardens were gradually transformed into high-value informal community plantings (social, cultural, or economic and environmental value). For example, guerrilla gardening in Europe and America [37,38,39], where residents illegally occupy public or private vacant and abandoned land in their communities without planning approval for planting, may damage the urban environment and the quality of green space [40,41]. However, some guerrilla gardens with social, economic, and environmental values have been approved as community gardens for the use of the residents after a consensus by the municipality [42]. In some countries, community gardens have also evolved from private spaces for small groups to legal public open spaces for all, such as the City of Sydney, which encourages residents to organize themselves to manage, build, and operate community gardens, and empowers them as legitimate open spaces that can stand on their own. [43]. In other developing countries, including Brazil, the Philippines, and Africa, high-value informal community growing has gradually gained the support of local governments and has become an effective vehicle for environmental improvement in urban communities [22,44].

In many Chinese cities, high-value informal community plantings are preserved as community gardens for residents to share. The phenomenon is now spread across all cities in China, such as Shanghai, Beijing, Guangdong, Zhejiang, Hunan, Sichuan, and other regions [7]. Relevant rules are formulated by the neighborhood committee (Community Management Organization) with the purpose of regulating residents’ planting behavior, creating a good community environment together, consciously optimizing landscape design, and community residents working together to maintain and create orderly planting activities. Therefore, it is important to study informal community planting by judging attitudes of residents and investigating its current characteristics, which can provide a reference for the governance of informal community planting in China.

Current research on informal community growing in China has focused on the current horticultural environment, the types and distribution of fruits and vegetables grown, the composition of growers, the motivation for planting, and the attitudes of concerned residents. He et al., conducted a research study on informal community growing among residents of an emerging settlement in Hangzhou, China, and found that the environment of cultivation was more private, the planting was mainly daily edible vegetables, the cultivation population was mainly over 50 years old with a middle income of 3000–5000 RMB/month having lived in the city for more than 10 years, and that the residents chose cultivation because of personal preference, abundant leisure time, food safety, social interaction, and landscape shaping. They were accepted by more than half of the surrounding non-growers. The satisfaction of both growers and the surrounding non-growers was high, and it was recommended that residents be allowed to grow without damaging the public environment and under the organization of community committees [6]. Yuan et al. studied the function, spatial distribution, planting forms and types, and attitudes of residents in informal community planting in an emerging residential area of Kunming, China. A small number of them were in public green areas and sidewalks, the planting forms were recycled containers and ground planting, and the planting types were mainly daily spices and vegetables [8]. It was proposed that some informal community planting should be incorporated into the urban planning system to become urban agriculture.

The aforementioned related studies proposed governance measures for informal planting by analyzing the current characteristics of informal community growing, the composition of the relevant population, the satisfaction of growers and affected non-growers, and other factors in emerging settlements. However, informal planting is not only found in emerging settlements, but is also prevalent in other urban spaces, especially in the mountainous cities of southwest China [45]. The author visited several areas in Chongqing, a mountainous city in southwest China (Changshou district, Wanzhou district, Qijiang district, and Yongchuan district), and found that the presence of informal community planting is extremely common, and residents usually plant on slopes and unused spaces above 25 degrees that are difficult to build on. These spaces are located throughout the city, especially in urban community parks and square slopes, sloping community public green spaces, and unused community edge slope spaces, where informal community planting may differ from the current situation and residents’ attitude in emerging settlements. There is a lack of research on the basics of informal community planting in mountainous cities in southwest China. Therefore, this paper attempts to study the basic characteristics of informal community planting and residents’ satisfaction in mountainous cities to provide suggestions for the governance of informal community planting.

Based on practical studies and surveys, this study selected a representative informal community growing area in Yongchuan District, Chongqing, for investigation and research, with the aim of:(1)Exploring the basic characteristics of informal community planting in mountainous urban areas (characterized by the planting scale, topographical features of the cultivation site, food types, distribution, and planting methods).(2)Understanding the structure of growers and growing motivation through questionnaire interviews and statistical analysis, and to understand the perceptions of affected non-growers on informal community planting.(3)Assessing the satisfaction of residents (growers and surrounding affected non-growers) with the current situation of informal community growing through semantic differential questionnaires. The assessment of residents’ satisfaction with informal community growing requires the selection of component factors, including various social, environmental, and economic benefits, alongside increased food supply [22,46]. The cultivation process and satisfaction both increase when the benefit is higher. Moreover, the benefits associated with the two groups are obtained differently, and the specific factor composition needs to be determined according to the reasons for cultivation by the growers and the perception of the surrounding affected non-growers.

Based on the above-mentioned study, results of informal community planting characteristics and residents’ satisfaction in mountainous areas, and governance recommendations for informal community planting in mountainous areas, are presented in the discussion. This study can also be used as a reference for other mountainous regions in China or other similar countries for the renewal of informal community farming.

## 2. Research Methods

### 2.1. Study Framework

The study was divided into 3 phases (Figure 1). The first phase was study area selection, which needed to meet the following requirements: (1) informal community growing covers a large area and uses sloping land that is difficult to build on. The study of the characteristics of planting in an area susceptible to erosion and soil quality damage will reveal the planting measures used by the population to prevent this problem and highlight the basic situation of informal planting in the mountains; (2) the study area is surrounded by a dense population and planted with a diverse demographic composition, covering almost all types of dwellers; and (3) the study area is located in the center of the city, where land value is high and is less likely to be planted than the distant suburbs. 

The second phase was data collection, which included basic characteristics of informal community farming in the sample area (basic data on the natural topography of the sample area, food types and distribution, and planting methods), basic information about the respondents, reasons for planting, perceptions of non-growers, and satisfaction of residents (growers and non-growers) with informal community farming.

The third phase was data analysis, which contained spatial analysis of the natural environment and farming status of the site through GIS and statistical analysis of the questionnaire results through SPSS23.

### 2.2. Study Area

The study area was selected from Yongchuan, Chongqing in southwest China. Chongqing is one of the most developed cities in China and belongs to the center of the Chengdu-Chongqing Economic Circle. Chongqing is a typical mountainous city with undulating topography and many slopes that are difficult to build on [47], resulting in the prevalence of vacant and abandoned land left behind. Through visits to Chongqing’s main city and some districts, it was found that sloping land and vacant and abandoned land were used by residents for growing (Figure 2). 

Yongchuan district is located in the western part of Chongqing, which is a sub-center city of Chongqing. Since the beginning of 2014, the growth rate of the population over 60 years of age has increased, while residents in other age groups have maintained a downward trend. From the data, it is clear that Yongchuan is a typical aging society [48]. As a result of rapid urbanization, cities are expanding outward and taking up rural land, resulting in an influx of rural residents into the cities [49], most of whom were once engaged in growing. As a result, informal community growing in Yongchuan has become increasingly common.

The built-up area of Yongchuan is 68.3 square kilometers, and informal community growing is common. According to the statistical analysis of the site data provided by the government, the area of informal community planting in urban areas is approximately 384.4 ha (949.8 acres), accounting for 5.6% of the built-up area of the city. The study area selected in Yongchuan forms only a part of it, approximately 46.93 hectares (74.55 acres are informal community planting sites) (Figure 3). The study area was originally rural land and was later zoned as an urban park. Some areas have been built into urban parks and squares for leisure, and many sloping areas have been reserved for difficult construction. Due to the lack of management in the study area, residents around the area have used abandoned sloping land to plant, thus forming a large area of vegetable fields and fruit forests. The case area is a large area planted by the informal community of Yongchuan (approximately 7.85%).

By visiting the residential areas around the 500 m range of the study area (the probability of planting by the residents in this range is higher), it was found that various communities exist in the surrounding area, including self-built, relocated, and affordable housing communities, and normal and high-end commercial housing communities. The number of existing neighborhoods within the range is counted as 2778 households with a diverse population structure. Therefore, it is clear from the scale, characteristics, demographic features, and location of informal community cultivation that the area is representative.

### 2.3. Data Collection

We visited local authorities to obtain related basic data, including land use and topography, etc. Field research was undertaken to obtain basic information on planting species, distribution, and planting methods in the study area. The basic information of the interviewees and their satisfaction with informal community growing in the region were conducted through questionnaires. The questionnaires were divided into 2 categories based on the above. The first general questionnaire investigated the interviewees’ age, gender, occupation, income, household registration (Hukou), education, farming experience, reasons for cultivation, and perceptions of non-growers. The second questionnaire, based on the semantic differential method, investigated the satisfaction of growers and affected non-growers.

The survey covered the period from September 2020 to December 2020, with further supplementary research information from April 2021 and March 2022. A total of 500 questionnaires were distributed and 465 were collected, with a recovery rate of 93%. A total of 440 questionnaires were valid, with an effective rate of 94.6%. Among them, 270 questionnaires were distributed to growers (250 of which were valid with an effective rate of 92.6%) and 230 questionnaires were distributed to non-growers, 190 of which were valid (198 were originally collected, of which 8 were unfamiliar with informal planting in the study area and were not counted), with an effective rate of 82.6%, and all of which were greater than 70%. The interviewees were all residents within 500 m of the study area and closely associated with it.

### 2.4. Data Spatial and Statistical Analysis

Firstly, the basic characteristics of cultivation in the sample area were analyzed. The planting scale and topography of the site were analyzed according to the data provided by the government, the types of cultivation and their distribution characteristics were analyzed through field visits, and the cultivation methods were tracked. The basic information of respondents was analyzed by descriptive statistics using SPSS23 to determine the composition of growers, the reasons for cultivation, and the perception of non-growers.

Secondly, the composition of the satisfaction factors of residents was determined. According to related studies, it is known that the satisfaction of growers and non-growers consists of 5 components: food access, social, economic, and environmental benefits [22,46], and planting process. However, the planting characteristics, composition of growers, and motivation for planting in informal communities in mountainous areas differed significantly from other studied regions, as did the composition of specific factors in the 5 major factors. Moreover, growers and non-growers have disparate benefic orientations, and the factors that constitute satisfaction are also different. Therefore, it is necessary to further select relevant factors based on the reasons for cultivation by growers and the perceptions of non-growers (the reasons for cultivation by growers and the perceptions of non-growers have been determined by questionnaire statistics in a later section), and the selection of factors for the 5 major factors is shown in Table 1.

Thirdly, the SPSS23 tool was used to determine the reliability and validity of questionnaires.

(1). Reliability test: Cronbach’s alpha coefficient (confidence coefficient) was calculated by reliability analysis with the following formula:(1)$$a=\frac{K}{K-1}(1-\frac{\sum_{i=1}^{K}S_{i}^{2}}{S_{t}^{2}})$$

‘a’ represents the reliability coefficient, ‘K’ denotes the number of test questions, $\sum_{i=1}^{K}S_{i}^{2}$ is the variance of the score for question ‘i’, and $S_{t}^{2}$ is the variance of the total score obtained for all questions. The reliability is between 0 and 1. The closer to 1, the better the reliability coefficient and the higher the internal factor reliability.

(2). Validity test: the Kaiser–Meyer–Olkin (KMO) value and Bartlett’s test of questionnaire factor validity by principal component analysis were used, and the KMO value was calculated as follows:(2)$$KMO=\frac{\sum \sum_{i\neq j}r_{ij}^{2}}{\sum \sum_{i\neq j}^{2}r_{ij}^{2}+\sum \sum_{i\neq j}a_{ij}^{2}}$$

$“r_{ij}^{2}”$ denotes simple correlation coefficient, $a_{ij}^{2}$ denotes partial correlation coefficient, when $a_{ij}^{2}\approx 0_{\;}^{\;}$, KMO ≈ 1, when $a_{ij}^{2}\approx 1_{\;}^{\;}$, KMO ≈ 0, and the KMO value is between 0 and 1. When the KMO value is greater than 0.70, it is acceptable and indicates a good relationship between the factors; when the KMO value is less than 0.50, the data is not suitable for factor analysis [50].

Fourthly, the semantic differential (SD) method was used to calculate the satisfaction of residents.

The SD method was first used in psychological research in 1957 by Charles E. Osgood in his book [51] to measure psychological feelings through verbal scales. Compared with traditional methods, the SD method can reflect the real demands of space users more prominently and has stronger applicability. Through this method, quantitative data on the respondents’ feeling constructs can be obtained to accurately determine the respondents’ satisfaction with something [52]. The specific operation includes 3 steps. The first step is drawing up the rating scale and collecting the respondents’ psychological feelings and actual experiences about the characteristics of the spatial environment. The second step is quantifying their feelings by dividing the survey factors into 5 categories: very satisfied (2 points), satisfied (1 point), general satisfied (0 points), dissatisfied (−1 point), and very dissatisfied (−2 points). The third step is calculating the overall score of each factor according to the proportion of residents’ choices. The formula for calculating the composite score is as follows:
(3)$$S=\frac{\sum_{i=1}^{n}{ [2\times P_{1}+P_{2}+P_{4}+2\times P_{5}]}_{i}^{2}}{n}$$

‘S’ denotes the composite score of each factor, ‘P’ refers to the percentage of each factor on different satisfaction options (P_1–5_ is the proportion of people on the 5 options), and ‘n’ is the number of factors.

Finally, the test of variance of residents’ satisfaction was acquired.

The test of variance refers to the study of differences in different dimensions of variables through tests such as independent sample *t*-test, Chi-square test, and one-way ANOV. In this study, only independent sample *t*-test and one-way ANOVA were applied according to the characteristics of the data. When the significance test is less than 0.05, it indicates that different variables have differences in satisfaction, and vice versa. Satisfaction of each variable was compared by a multiple comparisons test [50].

## 3. Results

### 3.1. Basic Characteristics of Informal Growing

The characteristics of informal community planting in mountainous areas were revealed by investigating the planting scale, topographical features, food types, distribution, and planting methods in the study area. From land use data, it is known that the study area has approximately 74.55 acres of informal planting land, accounting for 64.3% of the entire area (Figure 3), and approximately 0.027 acres of informal planting land per household (109.27 m^2^ per household), which can fully supply the food needs of each household. The topography of the study area gradually declines from east to west, and the overall landscape is concave and undulating, with a relative height difference of 64.5 m. The internally cultivated vegetable plots are distributed in irregular shapes, following the mountain and along the topography (Figure 4). In such a mountainous environment, the study area has only one dirt and gravel road of 0.8 m width, and the others are all muddy roads of 0.5 m width. It showed that the site is in a harsh environment and there is a lack of infrastructure. Residents choose crops to cultivate according to the terrain, and plant crops with well-developed root systems, deeper soil penetration, and strong soil-fixing abilities in flat areas, such as corn, potatoes, tomatoes, eggplants, and shallow-rooted crops, such as green onions and lettuce (Figure 5). The growers divide the fruit and vegetable species into two major parts: daily consumption and local foods (Table 2). Daily consumption foods are usually grown in large areas in a faceted layout, with an area of 20–30 square meters. Local food is interspersed in the daily food, in a dotted layout, with an area of 2–5 square meters. The study area is a mountainous terrain landscape, which is difficult to cultivate and does not allow the use of convenient mechanized tools. The planting methods consist of a series of traditional planting processes, such as seed spreading and seedling transfer, fertilization (composting), soil loosening, watering, harvesting, and waste disposal (Figure 6).

It is obvious that the phenomenon of large-scale spontaneous cultivation in urban centers is similar to the reuse of abandoned land by residents in some countries to form collective farming [24]. This large-scale collective planting also indirectly reflects the demand of the inhabitants and the wisdom of using the site to obtain a variety of food and traditional farming methods to reduce inputs and increase crop yields. It also reflects the difficulty of the growing process.

### 3.2. Basic Information of Interviewees

A total of 200 (45.5%) males and 240 (54.5%) females were interviewed in this study, 73% of whom are over 51 years of age, with a predominance of elderly people. In terms of household registration (Hukou), most of the growers have moved to live in the city for a short period of time (only 6.8% are urban residents who have lived for more than 10 years), and most have rich experience in farming (Table 3). In contrast, the non-growers have lived in the city for a relatively long time (the highest percentage of residents (35.3%) who have lived there for more than 10 years). In terms of occupation and income, 78.4% of the growers were unemployed or retired, while small percentages were engaged in business (10.8%) and labor (10.4%). A total of 74.8% of the growers earned less than 1700 RMB per month (the minimum wage in Chongqing is 1700 RMB), and nearly half of them had only received elementary education (38.8%) or less (18.8%). In contrast, non-growers generally have stable occupations and incomes (most earn more than $1700), and most non-growers are educated above secondary school/junior high school, which is a higher level of education than growers (Table 4).

The results show that the demographic composition of the growers is dominated by socially disadvantaged groups (low income, old age, joblessness, short migration to the city), for whom perhaps growing is an indispensable part of life.

### 3.3. Reasons for Planting

The questionnaire revealed that the reasons for residents to plant included seven aspects (Figure 7a): spending time (40.4%); decoration of idle plot (44.4%); social interaction (4.8%); economic gain (64.8%); food safety (58.4%); exercise (31.6%); and preference (5.2%). It is apparent that economic gain is the main reason for cultivation, and as the above-mentioned basic information statistics on the growers show, the general income of the growers is low, reflecting the fact that cultivation perhaps becomes an important part of the occupation and income for them. Surprisingly, very few people plant for the purpose of social interaction, hobbies, and physical exercise, which is significantly different from the reasons for planting among residents of other regions.

### 3.4. Perception of Non-Growers

The survey on the perceptions of non-growers showed that cultivation brings both positive and negative impacts (Figure 7b). Positive impacts include: access to a rural landscape (8.4%), food prices (36.3%), access to green food (28.9%), learning new farming skills (8.4%), and socialization and recreation (8.9%), while negative impacts include: smoke from burning (11.1%), mosquitoes (19.5%), odors (12.6%), landscape destruction (16.8%), and invading public resources (27.9%), etc. The results indicate that non-growers’ perceptions of the informal planting process are largely similar to other related studies. The learning of planting skills is unique to this area, however, probably due to traditional planting characteristics in the mountains. This allows residents who have no planting experience to learn traditional farming methods, which, combined the vast area of cultivation, creates a rural landscape similar to the countryside.

### 3.5. Results of Resident Satisfaction Analysis

#### 3.5.1. Questionnaire Reliability and Validity Test

The test was conducted through SPSS23, and the reliability and validity analyses utilized reliability analysis and exploratory factor analysis. The results yielded Cronbach’s alpha grower and non-grower satisfaction questionnaires of 0.724 and 0.737, respectively, indicating reliable internal factor consistency (Table 5). According to the results of the exploratory factor analysis in the table below, the KMO values for growers and non-growers are 0.806 and 0.754, respectively, both with significance *p*-values infinitely close to zero, indicating that the variables are well related and suitable for factor analysis.

#### 3.5.2. Resident Satisfaction Based on the SD Method

According to the questionnaire research of the SD method, the evaluation results of each factor were obtained by analysis in SPSS23 (Table 6 and Table 7), showing that the overall satisfaction ratings for growers and non-growers are 0.14 and 0.02, respectively, which is moderately low. In terms of individual factors, both showed higher satisfaction with food quality in terms of food access, with scores of 1.16 and 0.94, respectively, while the performance in terms of food availability and abundance was completely opposite, with the former being more satisfied (1.25, 0.41) and the latter being average (0.22, −0.29). It can be inferred that the varieties sold may also only be for larger yields of vegetables and most types of food, which growers still consume on their own, reflecting the passion and demand of the non-growers for planting foods. The economic benefits were more satisfactory for both (0.91, 0.95). In terms of social benefits, both showed dissatisfaction with social interaction (−0.41, −0.95). In fact, we can observe from the satisfaction scores of growers with interference from the outside (−0.23) that due to their spontaneity and individuality, each one considers only his own interests and is prone to conflicts and contradictions. In terms of the cultivation process, apart from psychological satisfaction (0.58), the growers were very dissatisfied with the exercise benefits (−0.40), safety (−0.65), and convenience (−0.55). Perhaps instead of exercising, cultivation in mountainous areas has become a physically demanding agricultural activity with certain dangers. Moreover, non-growers also showed dissatisfaction due to the negative effects of the planting process. In terms of environmental benefits, both were the least satisfied, as not only did planting not improve the landscape environment, but it also damaged the public environment due to the unrestrained nature of planting.

#### 3.5.3. Differential Test of Resident Satisfaction

An independent samples *t*-test showed no significant difference in gender (0.740 (growers) and 0.20 (non-growers), much higher than 0.05). According to the results of the one-way ANOVA, it is clear that the two satisfaction levels are different in other variables, and the significance is distinctively less than 0.05 (Table 8).

According to the results of multiple comparisons, it is obvious that those who have lived in the city for less time, older individuals, those with lower incomes and a lower level of education, and those with more experience in planting get higher satisfaction from growing.

From the results, it can be inferred that residents with high satisfaction levels belong to socially disadvantaged groups. The possible reasons for this are threefold: first, older individuals who have moved to the city for a short period of time after experiencing rural farming life may still be accustomed to the rural lifestyle and may find it difficult to quickly integrate into urban life, preferring to adopt the agricultural lifestyle. Second, due to joblessness and low income, they only focus on the income brought by farming and pay less attention to other negative factors. Third, the lack of education also indirectly leads to the group’s lack of cognitive ability to consider the multiple values of farming, so they will be easily satisfied with the current spatial environment. The group with low satisfaction levels generally have stable jobs, relatively high income and education levels, and have lived in the city for a long time. This group is relatively more concerned about the quality of life and they struggle to accept the negative impacts of cultivation on the environment. They are also less concerned about the economic gain brought by cultivation and are more likely to be oriented by hobbies, study, physical exercise, recreation, and other purposes.

## 4. Discussion

### 4.1. Potential for Transformation of Informal Community Planting

Informal community planting in mountainous cities arises for two reasons. On the one hand, the lack of management of community public slopes, construction difficulties, and delay in the construction of parks encourages planting. On the other hand, the lack of attention to the needs of vulnerable groups, which inevitably leads to a series of behaviors to defend their own interests [53,54], such as informal community planting. The above reasons are basically the same as other cities [6,9]. However, informal community planting in mountainous areas is different from emerging settlements in terms of basic characteristics, planting scale, the topographical features, food types, distribution, and planting methods in the study area, and composition of growers, motivation for cultivation, and perceptions of non-growers (Table 9).

Informal community planting may be detrimental to the public interest (a series of negative impacts on the environment) and should be discouraged. However, from basic characteristics of informal growing in mountainous urban areas it is undeniable that residents take full advantage of the topography to selectively grow crops for a diversity of foods. They use traditional farming methods to reduce the cost of farming inputs and increase crop yields, coupled with a scale so large that it can provide food for the community (0.027 acres per household) and meet the needs of vulnerable populations. There are also many informal community plantings (949.8 acres) in Yongchuan District, Chongqing, (Figure 3), which if integrated into community gardens or urban agriculture oriented to food supply, would greatly reduce dependence on external food and promote localized food production [55]. Moreover, many regions are actively building community gardens for the socially disadvantaged [56,57,58]. The prevalence of informal community planting in the mountains may in fact provide the basis for the future establishment of food networks and social security functions.

If it is to be optimized and managed by local governments, it will require some economic investment. We can know from the survey statistics that most growers accept the management of the local government and are willing to pay a certain amount to keep planting (Figure 8). However, studies have shown that it is difficult to support the management and operation of community gardens by relying solely on residents to raise funds and manage them [26,43]. Through raising funds, government policies, and economic support, resident participation can make full use of its benefits [59]. There are already many community gardens that have been operated through partnerships with various groups and have proven their value [60]. Therefore, instead of just clearing the land, we should analyze and study its existing characteristics and develop targeted measures to take advantage of high-quality informal planting areas with social, cultural, economic, and environmental value in mountainous areas [24,26,29].

### 4.2. Residents Satisfaction

#### 4.2.1. The Pros and Cons of Informal Community Growing in Mountainous Urban Areas

Currently, urban food relies on outside suppliers with a single type of food and questionable food safety, making it difficult to meet residents’ demands. However, informal community growing in mountainous urban areas caters to residents’ demand for green food, becoming a supplement to market food and effectively reducing food miles and carbon emissions [61,62,63]. Planting also brings economic and psychological satisfaction to socially disadvantaged groups, relieving their loneliness and life stress, and reducing social problems, which is one of the reasons why community gardens are built in various countries [14]. Moreover, planting also caters to urban dwellers’ desire for an idyllic life, and it is convenient to learn about farming skills (0.32) which has some educational function [64].

However, cultivation in mountainous areas does present various difficulties, such as safety issues (−0.65), watering, and fertilization during cultivation (−0.55), which neither satisfy the need for physical exercise (−0.4) nor guarantee safety, as well as external disturbance due to the negative impact of planting on public resources (−0.23), making cultivation hindered.

Informal planting in many areas also faces the same set of problems mentioned above [65], but the problems caused by informal planting have been improved through landscape creation and public management [37]. This can organize the residents to invest in the construction of infrastructure and avoid negative effects by guiding the residents to plant, so as to improve the social and environmental benefits of the site [42]. Based on the characteristics of informal cultivation in mountainous areas, we have improved the safety protection and management of the cultivation process, balanced the interests of all parties of the resident groups, and used it to enhance the happiness of resident groups.

#### 4.2.2. Variability of Residents’ Demands

Generally, older individuals who have only lived in the city for a short time, those with lower levels of income and education, the unemployed, and those with more planting experience are more satisfied with informal community gardening.

Different social classes have very different demands for urban quality, and marginalized groups have lower demands for urban quality than other social classes [66]. However, due to China’s rapid urbanization, lower classes have been passively moving into the cities as a marginalized group. Most growers are socially disadvantaged people who do not have a high-quality urban environment (a clean and comfortable public space, a disciplined lifestyle, etc.). They need some way to meet their basic needs, such as access to daily food, in a city where prices are high [67]. Informal community growing is an expression of the needs of this marginalized group. Therefore, if informal planting is optimized, they may be most actively involved in building an informal growing population, as it can help them to quickly integrate into urban life, supply their occupation, increase their income, and enhance their wellbeing [19]. In the case of most non-growers, they may want informal community growing to be transformed into a beautiful public space for the community. The construction of informal growing space takes into account not only the food supply, economic and social benefits, but also focuses on the quality of community spaces to meet their needs. This reveals that it is difficult to balance the interests of various groups in a purely residential co-construction manner, and that government intervention and support are essential [68] to set up and manage a non-profit organization [69]. Therefore, as far as the different groups in society are concerned, their demands are different, and it is worth to clarifying the interests of all parties in the future.

### 4.3. Planning and Management Recommendations for Informal Community Growing

Based on the research presented in this paper, the following three recommendations are made:(1)First, people need to strengthen the management of barren slopes and abandoned land and speed up the construction and management of abandoned lands. Relevant leisure spaces can be built according to the demands of dwellers [1], such as the construction of agricultural-nature parks with the countryside landscape atmosphere and farming education value, sports parks to meet exercise needs, etc., which can satisfy the requirements of residents while avoiding the phenomenon of planting everywhere.(2)Second, for some dense areas of disadvantaged people who are aging and have moved to live in the city for a short period of time, local managers should legalize planting, and establish relevant management regulations for growing in different areas, such as central areas to support three-dimensional planting, windowsill planting, and rooftop planting, etc., [70,71]. In urban fringe areas, where land is in abundant supply, managers can lease future long-term unused land for which residents pay a suitable rent, and the government or community council will invest in the construction of the associated infrastructure and operating and management expenses. Moreover, community councils can provide regular training and create community agricultural farming-knowledge learning camps [43]. For those who continue to produce negative impacts in the process of cultivation, appropriate penalties, such as fines and land withdrawal, can be imposed according to management regulations, and therefore, the landscape ornamentation of vegetable plots can be appropriately enhanced [6].(3)Third, it is important to explore food diversity and food availability in different existing planting areas in mountainous urban areas, to investigate satisfaction from the surrounding residents, and to analyze the relevant values (cultural value, countryside landscape value, etc.) that the area possesses, in order to appropriately develop it into a multifunctional community garden or urban agriculture in the future, as an urban food system [9,23].

### 4.4. Limitation and Future Research

This study selected Yongchuan, a mountainous city in southwest China, as a research sample to complement the study of informal community growing in China. Through fieldwork, questionnaires, and interviews, nearly 20% of the area’s residents were visited, and more accurate and detailed survey data were obtained to make the study results more scientific. Targeted questionnaire content was developed for both growers and non-growers to effectively reflect the true satisfaction of different residents with informal community planting. In addition, the accuracy and scientific precision of this study is likely to be higher compared to previous studies.

However, there are still some limitations in this study. The study only selected a single sample, and although this sample is representative, it is still difficult to fully reflect the characteristics of planting and the satisfaction of residents in informal communities, which may vary in other regions. Hence, samples from several different regions should be selected. Furthermore, there may be other factors affecting the satisfaction of growers, such as the physical condition of growers, planting scale, the distribution of plantings (some planting plots are close to entrances and flat plots, which may generally have higher satisfaction), etc. The satisfaction of non-growers will also be influenced by the spatial relationship between their houses and the planting site (it is likely that the closer they are to the planting site, the more they are disturbed by the planting process, and the lower their satisfaction). The above limitations will be discussed in a future study.

## 5. Conclusions

During the rapid urbanization in the mountainous regions of southwest China, many barren slopes and hard-to-build-upon abandoned land has been planted by the surrounding residents. This paper refers to it as informal community growing, which has attracted the attention of city managers, while a series of cleanup measures have been taken to no avail. The study sought to explore the characteristics of informal community growing in mountainous areas and residents’ satisfaction with it, as well as to provide measures of governance. The results suggest that the characteristics of informal community growing in mountainous cities, grower composition, and the purpose of cultivation differed compared to other cities (Table 9). Furthermore, growers and non-growers are satisfied with the food access and economic benefits generated by informal planting, but not with its social, environmental, and farming processes. Growers are willing to be officially managed and paid. Whereas satisfaction manifests itself differently in different population compositions, the marginalized (shorter time living in the city, lower income, less education, more experience in planting, older age) are more satisfied with planting, while the rest of the society is less satisfied. Overall, informal community growing in mountainous cities has some positive values (for example, the function of food supply, and the livelihood needs of socially different groups being met). Therefore, it clearly has the basis to become a formal community garden or urban agricultural area, and enhanced public management and community council involvement may mitigate its negative effects and bring about positive effects. Instead of forcing clear-cutting, urban managers should take the initiative to intervene, respond to the demands of different residents, balance the interests of all groups, and formalize the “informality,” which may bring more significant values.

## Figures and Tables

**Figure 1 ijerph-19-15178-f001:**
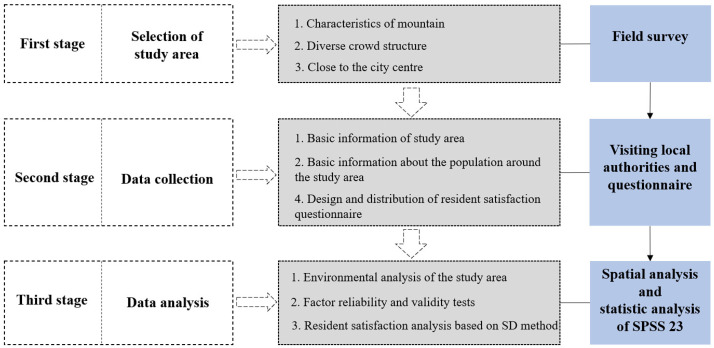
The workflow framework.

**Figure 2 ijerph-19-15178-f002:**
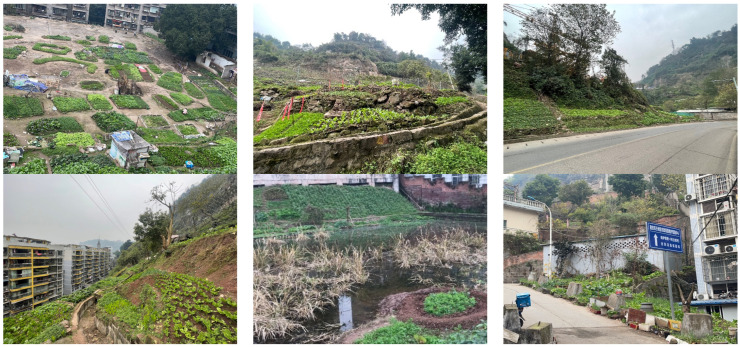
Various types of informal community growing phenomena in Chongqing. (Source: Canhui Cheng, 2021).

**Figure 3 ijerph-19-15178-f003:**
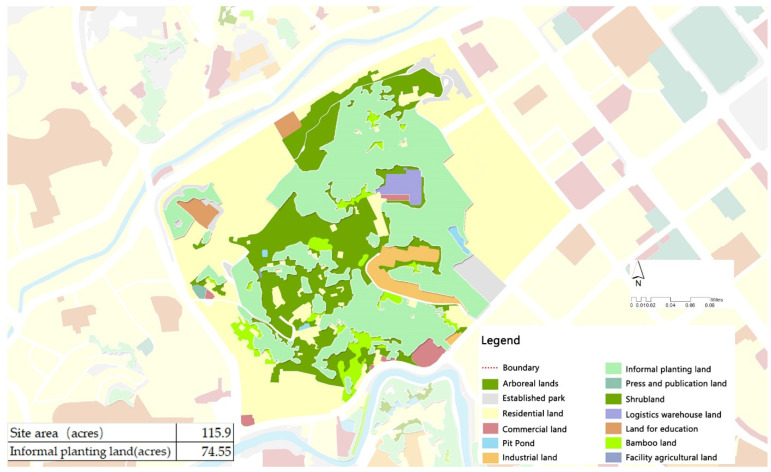
Statistics on the area of informal cultivated land in the study area. (Source: [48] District Government, 2022).

**Figure 4 ijerph-19-15178-f004:**
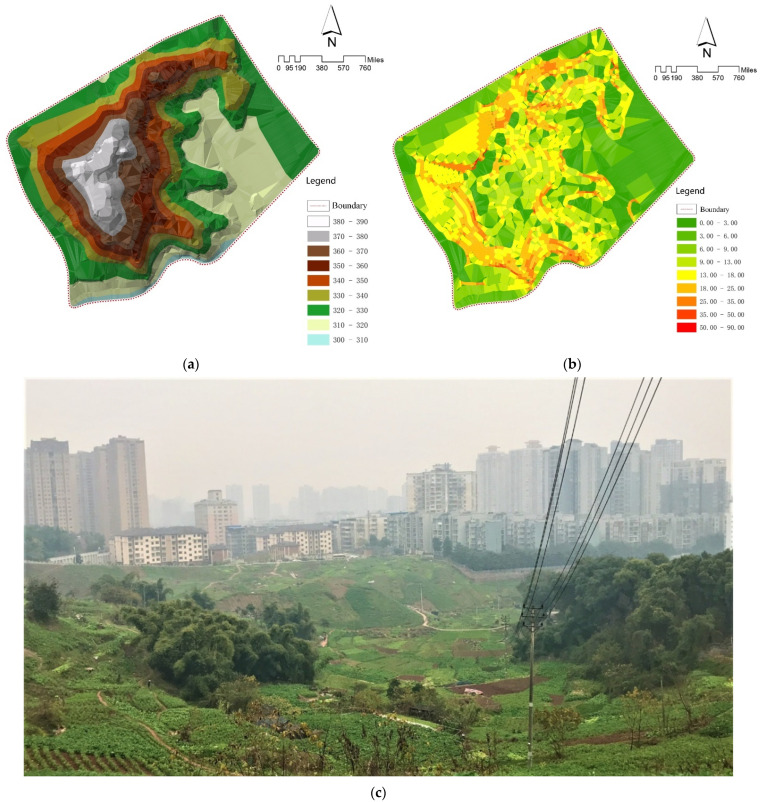
(**a**) Topography and geomorphology analysis; (**b**) site elevation and site slope; (**c**) site planting environment. (Source: [48] District Government, 2022 (**a**) and (**b**); Canhui Cheng, 2021 (**c**)).

**Figure 5 ijerph-19-15178-f005:**
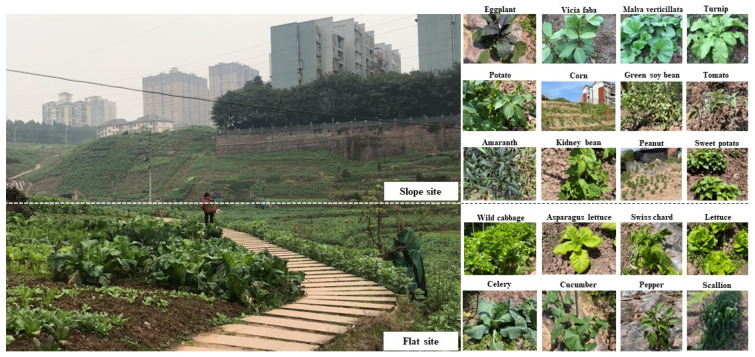
Fruits and vegetables grown on different slopes. (Source: Canhui Cheng, 2021).

**Figure 6 ijerph-19-15178-f006:**
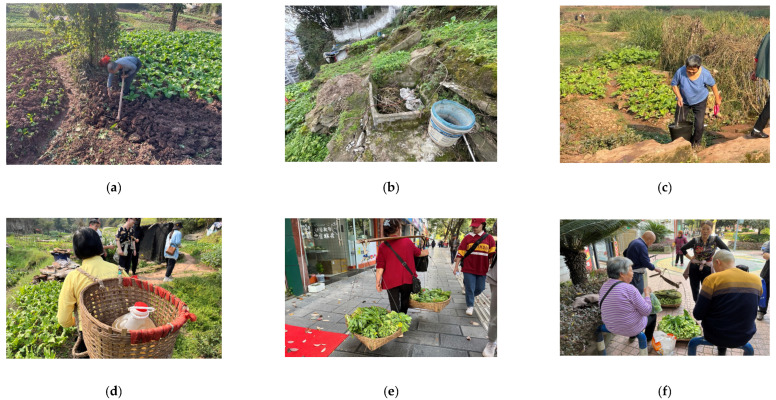
Farming methods in mountainous areas showing (**a**) seed sowing and soil loosening; (**b**) waste composting; (**c**,**d**) watering; (**e**) crop harvesting; (**f**) crop sales. (Source: Canhui Cheng, 2021).

**Figure 7 ijerph-19-15178-f007:**
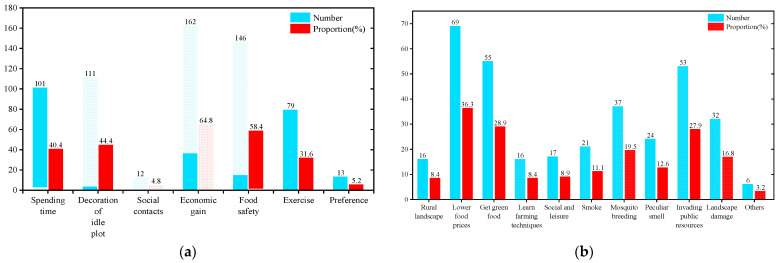
(**a**) Reasons for planting and (**b**) perception of non-growers.

**Figure 8 ijerph-19-15178-f008:**
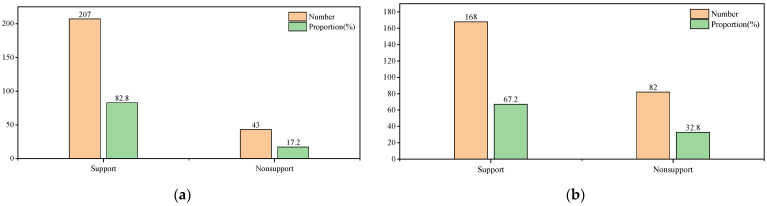
(**a**) Growers’ willingness to intervene in the management of community committees; (**b**) willingness to pay of growers.

**Table 1 ijerph-19-15178-t001:** Factor selection.

Satisfaction Factor	Growers	Non-Growers
Food Access	Food quality; food availability; food abundance	Food quality; food abundance; food availability
Economic benefits	Economic gain	Food prices
Environmental benefits	Planting on the landscape improvement effect; planting environment (sanitary environment, atmospheric environment, water environment, noise environment, soil environment, etc.)	Site environment (landscape environment, sanitary environment, pedestrian environment, etc.) and facility condition (recreational facilities, safety facilities, etc.)
Planting process	Safety of planting; convenience of planting (walkability, seeding, watering, fertilization and composting, loose soil, harvesting, waste disposal); exercise effect; psychological satisfaction	Negative impacts of the planting process (mosquito breeding, smoke from burning, odor generation, encroachment on public space)
Social benefit	Social interaction; external interference (complaints and warnings, theft and damage, conflict with others, suffering from cleanup)	Growing knowledge acquisition; social interaction

**Table 2 ijerph-19-15178-t002:** Types of fruits and vegetables planted.

Types	Specific Contents
Daily vegetable and fruits	Bok choy, spinach, kale, bok choy, garlic, ginger, onion seedlings, carrots, taro, rape, peas, lentils, potatoes, sweet potatoes, lettuce, mustard, thick-skinned vegetables, broad beans, winter sunflower, lotus root, white radish, cabbage, bamboo shoots, peppers, eggplant, fenugreek, sour mold, corn, yellow cauliflower, sesame, water vine lettuce, camelina, dates, tomatoes, cucumbers, citrus.
Local vegetable and fruits	Cauliflower, purple kale, lateral root, chrysanthemum, chrysanthemum, pepper, chrysanthemum, fern, mugwort, dumpling leaf, piper, sugar cane.

**Table 3 ijerph-19-15178-t003:** The basic information of growers.

Items	Contents	Number	Proportion (%)
Gender	Male	112	44.8
	Female	138	55.2
Age	≤30	7	2.8
	31–40	9	3.6
	41–50	31	12.4
	≥51	203	81.2
Registered Residence (hukou)	Urban residence (<5 years)	122	48.8
	Urban residence (5–10 years)	93	37.2
	Urban residence (>10 years)	17	6.8
	Rural residence	18	7.2
Career	Unemployed/farmers	196	78.4
	Businessmen	27	10.8
	Staff	26	10.4
	Other	1	0.4
Revenue	≤1700 RMB	187	74.8
	1701–4000 RMB	34	13.6
	4001–6000 RMB	23	9.2
	≥6001 RMB	6	2.4
Education	Uneducated	47	18.8
	Primary School	97	38.8
	Secondary School	79	31.6
	High School and above	27	10.8
Growing experience	<1 year	12	4.8
	1–3 years	35	14.0
	3–5 years	55	22.0
	>5 years	148	59.2

**Table 4 ijerph-19-15178-t004:** The basic information of non-growers.

Items	Contents	Number	Proportion (%)
Gender	Male	88	46.3
	Female	102	53.7
Age	≤30	19	10.0
	31–40	19	10.0
	41–50	34	17.9
	≥51	118	62.1
Registered Residence (hukou)	Urban residence (<5 years)	40	21.1
	Urban residence (5–10 years)	37	19.5
	Urban residence (>10 years)	67	35.3
	Rural residence	46	24.2
Career	Unemployed/farmers	43	22.6
	Businessmen	71	37.4
	Staff	59	31.1
	Other	17	8.9
Revenue	≤1700 RMB	71	37.4
	1701–4000 RMB	81	42.6
	4001–6000 RMB	21	11.1
	≥6001 RMB	17	8.9
Education	Uneducated	14	7.4
	Primary School	42	22.1
	Secondary School	79	41.6
	High School and above	55	28.9

**Table 5 ijerph-19-15178-t005:** Questionnaire reliability and validity tests.

Interviewees	Reliability Test			Validity Test		
	Cronbach’s Alpha	Cronbach’s Alpha Based on Standardized Terms	Numbers	KMO	Bartlett’s Spherical Test	*p*-Values
Growers	0.685	0.724	25	0.806	5697.08	0
Non-growers	0.74	0.737	17	0.754	1441.59	0

**Table 6 ijerph-19-15178-t006:** Results of grower satisfaction analysis based on the SD method.

Satisfaction Factor			Very Satisfied	Satisfied	Generally Satisfied	Dissatisfied	Very Dissatisfied	Score	Overall Score
Food access	Food quality		31.20%	60.40%	3.20%	3.60%	1.60%	1.16	0.94
	Food availability		34.80%	58.00%	4.80%	2.40%	0.00%	1.25	
	Food abundance		4.80%	43.60%	40.00%	11.20%	0.40%	0.41	
Economic benefits	Economic gain		22.00%	51.60%	22.40%	3.20%	0.80%	0.91	0.91
Environmental benefits	Planting on the landscape improvement effect		0.00%	0.80%	22.00%	59.60%	17.60%	−0.94	−0.45
	Planting environment	Sanitary environment	2.00%	5.20%	14.80%	67.20%	10.80%	−0.8	
		Atmospheric environment	20.40%	35.60%	42.80%	1.20%	0.00%	0.75	
		Water environment	1.20%	6.80%	8.80%	64.40%	18.80%	−0.93	
		Noise environment	4.80%	27.60%	38.80%	27.20%	1.60%	0.07	
		Soil environment	0.00%	2.40%	19.60%	66.80%	11.20%	−0.87	
Planting process	Safety of planting		6.00%	12.80%	12.00%	48.40%	20.80%	−0.65	−0.43
	Convenience of planting	Walkability	0.00%	6.40%	10.80%	35.60%	47.20%	−1.24	
		Seeding	8.40%	17.60%	49.20%	18.40%	6.40%	0.03	
		Watering	0.00%	7.60%	15.20%	68.40%	8.80%	−0.78	
		Fertilization and composting	0.40%	16.40%	21.20%	43.60%	18.40%	−0.63	
		Loose soil	2.00%	20.40%	69.60%	8.00%	0.00%	0.16	
		Harvesting	0.80%	9.20%	26.80%	50.00%	13.20%	−0.66	
		Waste disposal	1.20%	12.40%	16.00%	53.20%	17.20%	−0.73	
	Exercise effect		2.00%	21.60%	18.00%	51.60%	6.80%	−0.4	
	Psychological satisfaction		15.60%	36.80%	38.40%	8.80%	0.40%	0.58	
Social benefits	Social interaction		1.60%	7.60%	48.80%	32.40%	9.60%	−0.41	−0.26
	External interference	Complaints and warnings	3.20%	48.80%	24.40%	17.20%	6.40%	0.25	
		Theft and damage	2.40%	19.60%	29.20%	40.80%	8.00%	−0.32	
		Conflict with others	0.00%	2.40%	28.40%	39.60%	29.60%	−0.96	
		Suffering from cleanup	6.40%	31.60%	32.40%	27.20%	2.40%	0.12	

**Table 7 ijerph-19-15178-t007:** Results of non-grower satisfaction analysis based on the SD method.

Satisfaction Factor			Very Satisfied	Satisfied	Generally Satisfied	Dissatisfied	Very Dissatisfied	Score	Overall Score
Food access	Food quality	Food safety	38.40%	50.00%	5.10%	2.50%	0.00%	1.24	0.68
		Food freshness	27.80%	60.10%	3.00%	4.50%	0.50%	1.1	
		Food taste	34.80%	47.50%	10.60%	2.50%	0.50%	1.14	
	Food abundance		3.00%	15.20%	32.30%	40.40%	5.10%	−0.29	
	Food availability		10.60%	38.40%	11.10%	34.30%	1.50%	0.22	
Economic benefits	Food prices		21.70%	55.60%	16.20%	1.50%	1.00%	0.95	0.95
Environmental benefits	Landscape environment		0.00%	9.10%	14.60%	60.60%	11.60%	−0.75	−1.01
	Sanitary environment		3.50%	5.10%	16.70%	57.10%	13.60%	−0.72	
	Pedestrian environment		0.00%	1.50%	4.50%	60.60%	29.30%	−1.18	
	Recreational facilities		0.00%	2.00%	3.00%	66.70%	24.20%	−1.13	
	Safety facilities		0.00%	1.00%	1.00%	59.10%	34.80%	−1.28	
Planting process	Mosquito breeding		7.60%	18.20%	27.80%	28.80%	13.60%	−0.23	−0.2
	Smoke from the burning		1.00%	38.90%	41.90%	13.60%	0.50%	0.26	
	Odor generation		0.00%	2.50%	8.10%	65.20%	20.20%	−1.03	
	Encroachment on public space		10.60%	40.40%	10.10%	28.80%	6.10%	0.21	
Social benefit	Growing knowledge acquisition		4.00%	48.00%	21.70%	20.70%	1.50%	0.32	−0.32
	Social interaction		0.00%	4.00%	10.10%	64.10%	17.70%	−0.95	

**Table 8 ijerph-19-15178-t008:** Variability in resident satisfaction across variables.

Items		Code Name	Growers			Non-growers		
			Number	Significance Test	Multiple Comparisons Test	Number	Significance Test	Multiple Comparisons Test
Gender	Male	1	112	0.74	——	88	0.20	——
	Female	2	138			102		
Age	≤30	1	7	0.03	4 > 3	40	0	4 > 1; 1 > 3; 4 > 2; 2 > 3; 4 > 3;
	31–40	2	9			37		
	41–50	3	31			67		
	≥51	4	203			46		
Registered Residence (hukou)	Urban residence (<5 years)	1	122	0	4 > 1; 4 > 2; 4 > 3; 2 > 3; 1 > 2; 1 > 3	19	0.01	4 > 1; 3 > 1;
	Urban residence (5–10 years)	2	93			19		
	Urban residence (>10 years)	3	17			34		
	Rural residence	4	18			118		
Career	Jobless	1	196	0	1 > 2; 1 > 3	43	0	1 > 2; 1 > 3; 1 > 4; 4 > 2;
	Businessmen	2	27			71		
	Staff	3	26			59		
	Other	4	1			17		
Revenue	≤1700 RMB	1	187	0	1 > 2; 1 > 3; 1 > 4;	71	0	1 > 2; 1 > 3; 1 > 4; 2 > 3;
	1701–4000 RMB	2	34			81		
	4001–6000 RMB	3	23			21		
	≥6001 RMB	4	6			17		
Education	Uneducated	1	47	0	1 > 2; 1 > 3; 1 > 4; 2 > 3; 2 > 4;	14	0	1 > 2; 1 > 3; 1 > 4; 2 > 3; 2 > 4; 4 > 3
	Primary School	2	97			42		
	Secondary School	3	79			79		
	High School and above	4	27			55		
Growing experience	<1 year	1	12	0	4 > 1; 4 > 2; 4 > 3;	——		
	1–3 years	2	35					
	3–5 years	3	55					
	>5 years	4	148					

**Table 9 ijerph-19-15178-t009:** Comparison with informal community planting in other cities.

Item	Informal Community Planting in Mountainous Urban Areas	Other Urban Informal Community Planting
Reasons for informal planting	First, community legacy of difficult-to-build public slopes and unused spaces, lack of management, and lagging park construction. Second, lack of attention to the needs of vulnerable groups.	
Space distribution	Large-scale, three-dimensional planting dominated by selection of deep-rooted and shallow-rooted crops according to slope	Small-scale two-dimensional cultivation is dominant, with no particular choice of crop type
Type of land use	Mainly public space, such as barren slopes and bare land where construction is difficult, public green space, riverbank mudflats, road slopes, etc., which lack management	Mainly private spaces, such as yards, balconies, windows, roofs, etc., with some more private public green spaces
Planting method and difficulty	Traditional planting methods with a certain level of skill and difficulty in cultivation	Simple planting, small-scale land combined with foam boxes, washbasins, plastic bottles, and other portable-container planting, planting relatively simple
Planting types	Spices, daily and local vegetables	Spices, daily vegetables
Scale and physical form	Large-scale cultivation, local food 2–5 m2 in a point layout, daily food 20–30 m2 in a faceted layout	Small-scale planting, all for daily food, planting area is 0.3–5 m2 in a dotted layout
Growing residents	The elderly population with low income, no job, and experience in farming who have moved to the city for less than five years	Low-income, cultivation-experienced elderly population with more than 10 years of urban living
Reasons for planting	Economic gains, food security	Preferences, food safety
Perception of the growing process by non-growers	Access to cheap green food saves money, but negative impact on the environment is unsatisfactory	

## Data Availability

The data presented in this study are available on request from the corresponding author.

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
