# Peer review of "Informal Community Growing Characteristics and the Satisfac-tion of Concerned Residents in Mountainous Urban Areas of Southwest China"

_ijerph, 2022, doi:10.3390/ijerph192215178_

Round 1

Reviewer 1 Report

This research builds on a vast body of research into urban planning outside of regulatory systems. It introduces a localized experience in China that is not often included in literature reviews on the subject. The perspective of the research project is valuable to the discipline.There is evidence of research and an effort to connect with the local and international literature on the subject.

I found the research very valuable however, some areas need improvement. To start, the research question is quite vague the title leans toward the satisfaction of residents but not sure that is the most relevant question since the title hints at the possibility of removal; what has not been explored in the research question is not the findings but is mentioned as an exploration on the conclusions. Therefore, I suggest removing any recommendation about permanence or removal from the text since you are not testing for that issue.

The paper's methodology is clear, and the findings are valuable in answering satisfaction questions. The physical and morphological exploration of the planning areas could be further refined and integrated better into the findings and conclusions.

I think in general, the paper overestimates the reach of its conclusions and needs to be clear about what can be inferred by the methodology and what are the opinions of the authors; what I found in some cases was not back up by the data. However, there is much about the official management of planting areas but little on your questionnaire and results to support such recommendations.

Maybe the most crucial missed opportunity was to link informality in cultivation as a form of resistance and community engagement against capitalist/state forces; much of the literature on urban gardening explores such an issue. Community gardening in the global North and South have emerged as a motivation factor for action. I think engaging with some of this literature, and the one you already have highlighted, can help the readers better understand this local context and the similarities and differences. Your results clearly divide on age and economic /educational levels. However, there is little discussion about class and marginalization; crucial issues when talking about informality play an essential role in this issue. Further exploration of the literature on urban informality (resistance, resilience, conflict, entrepreneurial) could serve to ground your result in more profound theoretical arguments.

Finally, the paper can be further copy-edited. Terms such as high-quality urban environments, and legacy lands, need to be explained to the paper audiences since it is not clear what those terms mean in the context of your paper.

Reviewer 2 Report

The question is quite interesting. However, the whole paper needs to be carefully reorganized before it can be published to the public.

1. There are too many figures in section 2. Charts and graphs should be carefully selected to convey useful information, not to prove that you have done the work.

(1) The Yongchuan district in Fig 3 (the first row, right one) has a different shape with the red boundary of the second row, left one. It made me confused.

(2) Fig 4 and Fig 5 can be omitted since you have already described them in the text.

(3) Although Fig 6 gave direct information that there are different groups living in the study area, it wasn’t used in the SPSS analysis to further discuss the living condition impact on the satisfactory result. So, I suggest you omit the figure.

(4) Likely, table 2 divided the type of fruits and vegetables planted, however, it has nothing to do with your questionnaire. Thus, an extra table is meaningless.

(5) Fig. 7 draw the informal community growing land and marked different parts with A, B, C, and D … However, the marks are not used in your text.

2. The order and presentation of factors in Table 1 and Table 6,7 should be completely consistent.

3. Table 9 is the core conclusion of your work, which can distinguish the situation in Yongchuan from other urban informal community growing.  More analysis and description shall be explored. Perhaps the facts/pictures of your fieldwork (Fig. 9 & Fig.10) should be moved to this part to show its specialty, rather than scattered here and there.

4. There are papers cited in your conclusion, especially from lines 546 to 550. Conclusions should be drawn from questionnaires and analysis, not from other papers. Also, the suggestions in the last section should be moved to the Discussion section.

There are also some typos.

l  Affiliation 1, Affiliation 2, …shall be deleted from lines 6 to 11.

l  The number in enclosed in brackets has no meaning here in line 83.

l  The land utilization data shall be replaced with land use data in line 302.

l  Fig.12 shall be Fig.11.
